# Role of Circulating Tumor DNA in Hematological Malignancy

**DOI:** 10.3390/cancers13092078

**Published:** 2021-04-25

**Authors:** Miho Ogawa, Kazuaki Yokoyama, Seiya Imoto, Arinobu Tojo

**Affiliations:** 1Division of Molecular Therapy, Advanced Clinical Research Center, The Institute of Medical Science, The University of Tokyo, Tokyo 108-8639, Japan; ogawa-miho@g.ecc.u-tokyo.ac.jp (M.O.); aritojo@g.ecc.u-tokyo.ac.jp (A.T.); 2Department of Hematology/Oncology, Research Hospital, The Institute of Medical Science, The University of Tokyo, Tokyo 108-8639, Japan; 3Division of Health Medical Intelligence, Human Genome Center, The Institute of Medical Science, The University of Tokyo, Tokyo 108-8639, Japan; imoto@ims.u-tokyo.ac.jp

**Keywords:** liquid biopsy, circulating tumor DNA, ctDNA, MRD monitoring, malignancy

## Abstract

**Simple Summary:**

In recent years, next-generation sequencing has become a major tool in the management of cancer, advancing the diagnosis and treatment of hematological malignancies. However, the gold standard for cancer diagnosis and monitoring still involves invasive and painful procedures, such as tissue and bone marrow biopsies. These procedures involve physical risks, and a single biopsy cannot account for the spatial heterogeneity of tumors. The validity of circulating tumor DNA-mediated liquid biopsies has been receiving increasing attention. This review provides a brief overview of research on liquid biopsy in hematological malignancies, with special emphasis on circulating tumor DNA technologies, which may, in the near future, guide real-world decision making by hematologists.

**Abstract:**

With the recent advances in noninvasive approaches for cancer diagnosis and surveillance, the term “liquid biopsy” has become more familiar to clinicians, including hematologists. Liquid biopsy provides a variety of clinically useful genetic data. In this era of personalized medicine, genetic information is critical to early diagnosis, aiding risk stratification, directing therapeutic options, and monitoring disease relapse. The validity of circulating tumor DNA (ctDNA)-mediated liquid biopsies has received increasing attention. This review summarizes the current knowledge of liquid biopsy ctDNA in hematological malignancies, focusing on the feasibility, limitations, and key areas of clinical application. We also highlight recent advances in the minimal residual disease monitoring of leukemia using ctDNA. This article will be useful to those involved in the clinical practice of hematopoietic oncology.

## 1. Introduction

With the aging population on the increase, the global cancer incidence rate has been on the rise. Hematopoietic tumors are no exception, with one person diagnosed with blood cancer every three minutes and one person dying of it every nine minutes in the United States. Interest in the Precision Medicine Initiative announced in President Obama’s State of the Union address in January 2015 has spread worldwide. Precision medicine encompasses the prevention and treatment of diseases based on a detailed analysis of the genomic information and lifestyle of a patient and environmental factors. In cancer genome medicine, the practice of establishing a foundation for individual treatment based on a diagnosis guided by a comprehensive genetic analysis is being developed. However, invasive diagnostic methods are still widely used and remain an issue. This review summarizes the literature on circulating tumor DNA (ctDNA)-mediated liquid biopsy, a noninvasive and useful personal diagnostic approach based on genetic information. The clinical applications of ctDNA, which will be useful to those involved in the clinical management of hematopoietic tumors, are specifically emphasized.

## 2. What Is a Liquid Biopsy?

The term “biopsy” was coined by Besnier (France) in the late 1870s when conducting dermatological-related experiments. The word is derived from bio (life) and opsis (to see) [1]. Currently, biopsy is defined as “the removal of cells or tissues for examination by a pathologist” (https://www.cancer.gov/publications/dictionaries/cancer-terms/def/biopsy, accessed on 1 April 2021). There are various problems associated with traditional tissue biopsies. First, complications like pain and bleeding at the puncture site after biopsy are frequent [2,3,4]. Second, the detection of an early-stage tumor or residual lesions is unsatisfactory in tissue biopsy, and its application in determining treatment efficacy and prognosis is limited [5]. Further, it is difficult to capture the inherent molecular heterogeneity of metastatic tumors and the ability of cancer genomes to evolve dynamically based on a single local biopsy [6,7].

The term “liquid biopsy” was first used by Pantel et al. in their 2010 review of circulating tumor cells (CTCs) to refer to the clinical utility of peripheral blood sample analysis [8]. Now, in a broad sense, liquid biopsy indicates the isolation and analysis of tumor-derived materials (e.g., DNA, RNA, or even intact cells) from blood or other bodily fluids [9,10]. Samples for liquid biopsy include plasma/serum, urine [11], saliva [12], stool [13], and cerebrospinal fluid [14]. Some argue that cerebrospinal fluid does not qualify as a fluid for liquid biopsy because of the invasive nature of its collection.

## 3. Contents of a Liquid Biopsy

In a conventional biopsy, tissue from the lesion is collected for the morphological evaluation and quantification of nucleic acids (DNA, RNA), proteins, and metabolites contained in the cells. On the other hand, in liquid biopsy, CTCs, cell-free DNA/RNA, extracellular vesicles, and microRNAs (miRNAs) in the collected liquid are used to evaluate the characteristics of the origin cell. CTCs are thought to be part of cancer stem cells released into the bloodstream via the epithelial–mesenchymal transition and undergo hematogenous metastasis [15]. Therefore, analysis of the CTCs detected in the peripheral blood (PB) of cancer patients is useful for early cancer screening, elucidation of the metastatic process, prediction of prognosis, determination of therapeutic efficacy, and analysis of the mechanisms of therapeutic resistance [16,17,18,19,20,21]. There is a paucity of literature on CTCs in hematopoietic tumors, probably because, unlike solid tumors, the concept of metastasis is rarely used and because the detection of CTCs requires complex enrichment methods [22,23]. In addition, in multiple myeloma (MM), CTCs have been associated with the spread of extramedullary lesions [24,25,26,27] and have been applied in the genetic profiling of tumors [28].

In 1983, Johnstone and Harding discovered vesicles secreted by reticulocytes, and the vesicles were named exosomes in 1987 [29,30,31]. Since then, vesicles of different sizes and origins have, among others, been referred to as ectosomes, microvesicles, and shedding vesicles. These are now collectively referred to as extracellular vesicles (EVs). Their contents include proteins, nucleic acids (miRNAs and messenger RNAs [mRNAs]), lipids, and metabolites, and they have been shown to be a tool for intercellular communication [32,33,34]. In particular, miRNAs encoded by EVs are one of the bioactive molecules involved in tumor growth and drug resistance and have attracted attention in malignant lymphoma [35] and MM studies [36,37].

## 4. Cell-Free DNA and Circulating Tumor DNA

Cell-free DNA (cfDNA) refers to all non-encapsulated DNA in the bloodstream. In 1948, Mandel and Metais were the first to report the presence of cfDNA in the plasma of patients with systemic lupus erythematosus [38]. In healthy individuals, the concentration of cfDNA ranges from 0 to 100 ng/mL of blood (average ~ 30 ng/mL) [39]; the main sources are apoptotic or necrotic cells [40,41]. Normally, the DNA of apoptotic cells is rapidly degraded by DNase [42]. However, when the uptake of apoptotic bodies is impaired or when a large number of apoptotic cells are generated, such as following acute trauma [43], stroke [44], exercise [45], transplantation [46], infection [47], and cancer [48,49], the cfDNA concentration in plasma increases. In oncology, the tumor-derived fraction of cfDNA is known as ctDNA [50]. ctDNA can be used to analyze not only mutations but also methylation status, size fragment patterns, transcriptomics, and viral load [10,51]. Studies by Dennis Lo et al. regarding clearance time revealed that the circulating fetal DNA, after delivery, has a mean half-life of 16.3 min. The study suggested that tumor-derived DNA may be removed rapidly [52].

Diehl et al. sampled the plasma of colorectal cancer patients who underwent tumor resection and showed that the ctDNA levels determined before surgery varied widely and that the postoperative ctDNA half-life was 114 min. In the study, ctDNA levels reflected the total systemic tumor burden, in that, the levels decreased upon complete surgery and generally increased as new lesions became apparent upon radiological examination [49]. Bettegowda et al. evaluated the use of ctDNA to detect tumors in 640 patients with various cancer types. They found that detectable levels of ctDNA correlated with the stage of cancer; ctDNA was detected in 47% of patients with stage I cancer, 55% with stage II, 69% with stage III, and 82% with stage IV [53]. In a study using 3D volume reconstruction of computed tomography (CT) images, Parkinson et al. clearly demonstrated that the amount of ctDNA in the plasma of high-grade serous ovarian cancer patients reflects the degree of the tumor [54]. From these studies, ctDNA analysis is considered a real-time snapshot of disease burden.

Fan et al. measured the size of fetal cfDNA and found a dominant peak at ~162 bp and a minor peak at ~340 bp [55]. The size of cfDNA in cancer patients has also been reported to peak at ~180 bp, which is considered to be the result of protection from enzymatic degradation by histone binding to nuclear DNA during apoptosis. In contrast, DNA fragments larger than ~10,000 bp could originate from cells dying via necrosis [40].

For cfDNA sampling, plasma centrifuged from whole blood or commercially available blood collection tubes for cell-free DNA sampling is used [56]. In addition, the phenol–chloroform method, sodium iodide method, magnetic bead method, and commercial DNA isolation kits, are commonly used for extraction [57]. Depending on the experimental design, there may be concerns about sampling bias because of the use of samples that have been stored for a long time and the differences in collection methods among facilities.

The detection of ctDNA, which is present in peripheral blood at very low allele frequencies, requires high technology, and with the evolution of PCR methods (ASO-PCR, ddPCR) and the improvement of the data output of sequencers, highly sensitive detection methods have become feasible (Table 1). However, there is still no method that meets all the requirements, such as the cost and the number of facilities where it can be performed, for progress in clinical applicability [56,58].

Clinical trials are being conducted to use ctDNA to select treatments in solid tumors, and screen for cancer in healthy individuals [64,65,66,67,68,69]. In the case of hematopoietic tumors, there is a need to research cfDNA independently because the tissue collection method (hematopoietic tumor cells are contained in various tissues such as bone marrow, peripheral blood, and lymph nodes) and the genetic abnormalities observed in these tumors are different from those in solid tumors. Currently, ctDNA is being studied in hematopoietic tumors for the detection of tumor-specific mutations, evaluation of therapeutic efficacy, detection of minimal residual disease (MRD), prognosis prediction, and assessment of genomic heterogeneity. The application of cfDNA in hematopoietic tumors is described in this review.

## 5. Utility of ctDNA Characterization in Hematopoietic Tumors

### 5.1. Acute Myeloid Leukemia (AML)/Myelodysplastic Syndrome (MDS)

The study of cfDNA started again in 1994 [70] in AML/MDS, about half a century after the discovery of cfDNA in plasma by Mandel et al. In 2004, Rogers et al. used capillary electrophoresis to detect loss of heterozygosity in ctDNA from the plasma of patients with cytogenetically identified chromosomal abnormalities. They found that the plasma may be a potential substitute for bone marrow (BM) as a material for chromosome testing [71]. Although similar studies have not been conducted since then, the detection of chromosomal aberrations by ctDNA is expected to be re-examined using whole-genome sequences.

cfDNA extraction using off-the-shelf kits has become widespread since the 2000s (Table 2). In 2010, Gao et al. focused on the quality of cfDNA extracted using a QIAamp DNA Blood Kit (Qiagen, Hilden, Germany) and examined the plasma ctDNA integrity index of 60 acute leukemia patients using quantitative real-time PCR (qPCR) amplification of the β-actin gene. They concluded that plasma DNA integrity is incremental in acute leukemia and may serve as a biomarker for monitoring MRD [72]. This study became the basis for validating the usefulness of ctDNA in monitoring MRD in leukemia, which was later published in numerous publications. The ultimate goal of many researchers has shifted from conventional diagnostic approaches to using cfDNA-detected genetic mutations for diagnosis, treatment decisions, and prognosis. Iriyama et al. performed a global methylation analysis using bisulfite pyrosequencing based on the specific CpG sites of the *LINE-1* promoter and quantified a *TET2* mutation in DNA from plasma. The results showed that the methylation rate decreased rapidly in plasma ctDNA after azacitidine administration. Furthermore, in the quantification of the *TET2* mutant gene in ctDNA, they observed that the ratio of the mutant gene was almost at the same level as that in the BM CD34+/38− stem cell population. They compared their results with those of BM cells and PBMCs analyzed simultaneously and showed that ctDNA could be used for genetic/epigenetic analysis with higher sensitivity [73]. In 2015, Quan et al. quantified the copies of circulating nucleophosmin (*NPM*) mutations in plasma DNA of AML patients using qPCR and analyzed the association between ctDNA copies and clinical characterization. They showed that patients with high PB white blood cell and platelet counts and high BM blast rates had significantly high copy numbers of the circulating *NPM* mutant gene [74].

Hiseq2000 (Illumina) was introduced in 2010. With the development of next-generation sequencers, the target area for analyzing cfDNA mutations has also expanded rapidly. In 2016, Albitar et al. subjected the cfDNA of MDS patients to a panel genetic analysis targeting 14 genes. All samples were found to have at least one mutated gene, confirming the presence of an abnormal clone consistent with MDS. The authors noted that the diagnosis of MDS requires morphological and cytogenetic diagnoses but that neither of these can sometimes provide clear evidence and that molecular tests to detect abnormal clones have been relied upon in recent years [75]. NGS of cfDNA is expected to be a highly sensitive method to detect abnormal clones in MDS, where cytopenia may prevent the collection of sufficient peripheral blood nucleated cells. Suzuki et al., from the same research group as Iriyama et al., who performed ctDNA methylation analysis, evaluated the validity of targeted sequencing of cfDNA by comparing somatic mutations detected in MDS, BM DNA, and cfDNA. The Sanger method was used for most cases in this study; only two cases underwent NGS on BM DNA/cfDNA pairs; therefore, additional validation is needed [76]. cfDNA mutation analysis is now being validated using a panel targeting more genes; Zhao et al. performed cfDNA genetic analysis of MDS on a panel of 127 genes (Roche NimbleGen liquid phase hybrid capture chip). They concluded that ctDNA reflected genetic variation in BM DNA [77]. With the rise of digital PCR in the late 2010s, MRD monitoring studies targeting mutated genes became mainstream. Yeh et al. performed targeted sequencing using the BM of MDS patients treated with azacitidine and eltrombopag; here, detected driver mutations and karyotypic abnormalities were tracked in cfDNA using digital PCR during treatment. They demonstrated that serial monitoring of ctDNA allowed the concurrent tracking of both mutations and karyotypic abnormalities throughout therapy, and they were able to anticipate treatment failure. In addition, they revealed that ctDNA exhibited a differential response in the malignant subclones during therapy [78].

At the time of diagnosis of acute leukemia, 10^12^ leukemic cells were present in the patient’s body. Even after successful induction therapy and complete morphological remission with less than 5% leukemic cells in the BM on speculum, 10^9^ leukemic cells remained in the body. The number of leukemic cells below 10^9^ undetectable by light microscopy are termed as MRD and are considered important because they can proliferate and cause relapse after remission [79,80,81]. The MRD evaluation method used in clinical practice, qPCR, which detects genetic mutations in bone marrow DNA, has a high sensitivity of 0.1%. However, only patients with gene mutations detectable by commercial-based tests, such as APL (*PML-RALα*) and CML (*BCR-ABL*), can benefit from it. Currently, allogeneic hematopoietic stem cell transplantation (alloSCT) [82] is the only curative option for patients with high-risk or refractory AML and MDS. However, many post-alloSCT AML patients experience relapse and suffer another severe disease course [83,84]. In addition, AML patients with positive MRD pre- or post-transplantation have a higher risk of post-transplant recurrence and a lower survival rate [85]. In 2018 and 2019, we developed a ctDNA-based noninvasive MRD monitoring and prognosis system for patients with AML/MDS undergoing alloSCT [86,87]. We retrospectively performed whole-exome sequencing of tumor samples from AML/MDS patients (using the patient oral mucosa cells as a control) to identify driver mutations in the tumors, construct patient-specific ddPCR assays, and monitor MRD with ctDNA obtained from serum at 1 month and 3 months after transplantation (Figure 1). We found that patients who were MRD-positive at 1 month and 3 months after transplantation had an increased relapse rate 3 years after transplantation. We are currently conducting a prospective study of patients with AML/MDS undergoing alloSCT to verify the usefulness and feasibility of MRD monitoring using ctDNA. We will be able to report on this in the near future. In 2020, Nicholas et al. performed targeted NGS of ctDNA and BM DNA in patients with AML and suggested that they may be complementary to the assessment and monitoring of patients [88]. Zhong et al. detected monoclonal immunoglobulin gene rearrangement in AML ctDNA, which is expected to be applied in MRD monitoring [89]. The detection of IgH and TCR rearrangement is common in lymphoid malignancy but rare in AML. As we have shown, MRD monitoring by ddPCR has been used in many studies, but Christenson et al. did not consider it substantial. If the expression of SNPs in the wild-type gene sequence is not taken into account when creating a ddPCR assay, the correct allele frequency may not be detected [90]. In addition, the presence of clonal hematopoiesis with indeterminate potential (CHIP) should be noted in the evaluation of MRD by ctDNA. Clonal hematopoiesis is a condition in which a single or a very small number of clones maintain hematopoiesis. Mutations in epigenomic regulatory genes such as *DNMT3A*, *TET2*, and *ASXL1* are the most common causes of CHIP [91,92]. It has been reported that the majority of mutations detected in cfDNA of healthy individuals are CHIP, and if these are detected as mutations in ctDNA and subjected to MRD evaluation, the reliability of the test will be reduced [93]. There is an opinion that it is possible to exclude CHIP mutations by paired sequencing of leukocyte DNA and plasma cfDNA [94], but further validation is needed in terms of cost and time. It is hoped that MRD monitoring by ctDNA will be implemented in the clinical environment after repeated trial and error and sufficient validation.

The monitoring of ctDNA by continuous specimen collection is promising not only as a clinical test as described above but also as an evaluation tool in clinical trials to develop new therapeutic agents. Zeidan et al. performed ctDNA monitoring in a Phase Ib study of PLK1 inhibitor, onvansertib, for AML. They monitored VAFs with ctDNA before and after treatment with a gene mutation that was previously detected in the target sequence of tumor DNA. They concluded that clinical response to onvansertib could be predicted from changes in VAF after treatment [95]. This study did not have a large enough sample size to create a receiver operating characteristic curve, but as more such trials are conducted, the use of ctDNA as a test endpoint will be optimized.

### 5.2. Lymphoid Malignancy

The most frequently reported studies on ctDNA in hematopoietic tumors are in the field of lymphoid malignancy. In April 2021, we searched PubMed for “circulating tumor DNA, Leukemia, NOT review,” “circulating tumor DNA, Lymphoma, NOT review,” and “circulating tumor DNA, Multiple Myeloma, NOT review,” and found 159, 224, and 38 hits, respectively. This may be because lymphoma incidence is the highest among hematopoietic tumors (Cancer Facts & Figures, 2020; American Cancer Society, 2020).

In the case of myeloid malignancy, most studies have used ctDNA SNVs/Indels as detection targets. On the other hand, in lymphoid malignancy, immunoglobulin-heavy chain (IgH) rearrangement was initially emphasized. In 1997, using PCR, Frickhofen et al. detected clonal DNA from a rearranged IgH locus in the plasma samples of patients with non-Hodgkin lymphoma and acute B-precursor lymphoblastic leukemia [97]. Further, Zhong et al. detected IgH and T cell receptor γ gene rearrangements in plasma cfDNA from patients with non-Hodgkin lymphoma [98]. In 2011, He et al. detected specifically rearranged DNA fragments in patient plasma using IgCap, which captures and sequences the IgH genomic region [99]. Subsequently, the data output of NGS techniques increased, and high-throughput deep sequencing of immunoglobulins (IgHTS), a technique for the comprehensive analysis of CDR3 sequences of BCR genes by multiplex PCR, became available commercially. Armand et al. analyzed plasma ctDNA from 16 DLBCL and MLBCL cases using Sequenta Lympho SIGHT (Sequenta Inc., South San Francisco, CA, USA) and successfully detected IgH and IgK reconstitution [100]; Kurtz et al. compared ctDNA analysis with IgHTS and imaging diagnosis using PET-CT in 75 cases of DLBCL. They compared ctDNA analysis using IgHTS and imaging diagnosis with PET-CT [101]. In addition, Roschewski et al. argued the usefulness of ctDNA surveillance by IgHTS based on a 5-year post-treatment follow-up of 126 DLBCL cases [102]. Sarkozy et al. used IgHTS of follicular lymphoma (FL) cases to compare the chronotype of tumor DNA and plasma ctDNA and showed that the subclonal distribution between tumor and plasma was different in more than half the cases [103]. IgHTS of ctDNA for other diseases including HIV-related B cell lymphoma was performed by Wagner-Johnston et al. [104] and MCL by Kumar et al. [105]. For B cell malignancy, there are many reports, as mentioned above. Regarding T cell lymphoma, Zhang et al. performed T cell receptor HTS in 2021 and detected TCR rearrangement in 78% of the cases; hence, only a few reports exist [106].

Regarding the methods for detecting SNV/Indel, Hosny et al., in 2009, detected the *TP53* mutation in ctDNA of NHL patients using direct sequencing, but this method was limited to the target genes [107]. With the advent of CAPP-seq (cancer personalized profiling by deep sequencing), panel sequence, and low-pass WGS, the target gene region has been greatly expanded, and continuous monitoring by ddPCR as in myeloid malignancy has been reported [108,109,110,111,112]. CAPP-seq is an economical and ultrasensitive hybrid capture-based target sequence method developed by Newman et al. in 2014 [62]. WGS and WES have the advantage of measuring a wide range of genetic regions, but they have the disadvantage of high costs to achieve high detection sensitivity. However, CAPP-seq can detect four types of mutations (single-nucleotide polymorphism, insertion/deletion, copy number polymorphism, and fusion region) in cell-free tumor DNA with high efficiency by simultaneously sequencing cancer-specific mutated gene regions. Newman et al. at Stanford University performed CAPP-seq on ctDNA of NHL and showed better sensitivity than IgHTS in genotyping ctDNA [113,114]. This technique was applied to DLBCL by Rossi et al. [115] and classical Hodgkin lymphoma (cHL) by Spina et al. [116]. While CAPP-seq has the advantage of obtaining genotypic information with high sensitivity, the number of facilities that can perform CAPP-seq for hematological malignancy is limited.

A target sequence, which can analyze genetic variation in a specific genomic region, is widely used. Bohers et al. first published the detection of SNV/Indels in lymphoid malignancy [117]. This was followed by reports showing the evaluation of the validity of SNV/Indel detection in ctDNA [118,119,120], the association of genotyping-derived profiling with treatment response and prognosis [121,122,123,124,125,126,127,128], and the study of its applicability as a MRD monitoring material [129,130]. In DLBCL, Rushton et al. performed a 63-gene target sequence for ctDNA in 135 cases [125]. Camus et al. performed the first prospective ctDNA genotyping after cHL chemotherapy and showed that MRD detection by ctDNA was superior to that by PET-CT [129]. The target sequence is highly versatile because it is available commercially, but it is less sensitive in identifying mutations in cfDNA. Technical considerations, such as the expansion of the target gene region, are necessary. For CNS lymphoma, there are scattered reports on ctDNA in cerebrospinal fluid (CSF) [131,132,133,134], and some argue that ctDNA in plasma is not sensitive enough for detection [135,136]. The applicability of ctDNA in plasma should be carefully evaluated considering the characteristics of the disease.

Low-pass WGS can also detect fusion and CNV in addition to SNV/Indels. It has the advantage of providing richer information in samples with high tumor volume; Agarwal et al. used it to elucidate treatment strategies with ibrutinib and venetoclax [137].

Cytosine methylation in the CpG islands of gene promoter regions is another factor that has a significant impact on gene expression. In cancer cells, the expression of tumor suppressor genes is repressed by the aberrant methylation of CpG islands [138]. In the field of leukemia–lymphoma, methylation analysis of tumor suppressor genes has been actively pursued [139,140]. In 2003, Deligezer et al. suggested the presence of methylated tumor suppressor gene 16 in the cfDNA of patients with lymphoproliferative diseases [141]. In 2007, Shi et al. established CpG island DNA microarray methylation profiling in cancer; they detected methylation of the tumor suppressor candidate gene *DLC-1* in plasma DNA from non-Hodgkin lymphoma patients. They further reported that *DLC-1* methylation disappeared after response to chemotherapy as determined by quantitative methylation-specific PCR analysis [142]. Thus, DNA methylation analysis has been applied to the genetic characterization and monitoring of lymphoid malignancy.

Validation of ctDNA levels and integrity has also been active in lymphoma [143,144,145,146]. Delfau-Larue et al. quantified ctDNA using ddPCR and compared it to tumor volume measured by PET-CT, and showed that in FL, pre-treatment ctDNA levels correlated with total metabolic tumor volume measured by PET-CT and total tumor volume [147]. Kurtz et al. also developed the continuous individualized risk index (CIRI), a risk assessment scale that incorporates pre-treatment ctDNA levels for DLBCL divided into high and low [148]. A challenge for social implementation is to clarify the interpretation of ctDNA analysis results in combination with existing laboratory findings such as imaging, especially in lymphoma.

### 5.3. Multiple Myeloma

MM is characterized by monoclonal proliferation of plasma cells, the final differentiation stage of B cells. Most cases start as monoclonal gammopathy of undetermined significance (MGUS) and develop into smoldering multiple myeloma (SMM) with no clinical symptoms. Eventually, one of the four symptoms (CRAB) of hypercalcemia, renal failure, anemia, and bone lesions become associated with the diagnosis of MM [149]. At diagnosis, there is clonal diversity with the coexistence of dominant and minor subclones that have evolved from a common ancestral tumor-initiating cell or stem cell. At recurrence, clones are also heterogeneous and may be dominated by the same or different subclones from those at first appearance [150]. Tumors are multifocal in BM and secrete monoclonal immunoglobulins (M-proteins) in blood and urine. They are morphologically heterogeneous and have varying responses to therapy and tumor progression. Therefore, BM puncture at a single site alone is subject to sampling bias and yields only a limited molecular profile, which cannot reflect the diverse pathogenesis of various subclones. The measurement of cfDNA, which comprehensively reflects the entire tumor, is ideal for clarifying the pathogenesis. In particular, MRD monitoring, which has gained clinical significance with the development of new drugs [151], is expected to be applied to ctDNA as well. In this context, Sata et al. [152] and Oberle et al. [153] reported the detection of immunoglobulin rearrangements in ctDNA. Mazzotti et al. analyzed 47 cases of MM using the NGS MRD assay (IgHTS) from Adaptive Biotechnologies, claiming that MRD monitoring is not possible with NGS alone, which targets Ig gene rearrangement [154]. On the other hand, some argue that quantification of IgH rearrangements can be used for prognostic stratification [59,155].

For SNV/Indel, the analysis focuses on targeting *KRAS, NRAS,* and *BRAF* [156,157,158]. Mithraprabhu et al. analyzed cfDNA of MM patients for target gene mutations, including *KRAS*, *NRAS*, and *BRAF*, using ddPCR and continuously quantified and monitored ctDNA. They analyzed paired BM cell DNA and ctDNA from 33 relapsed or refractory MM patients and 15 newly diagnosed patients using targeted deep sequencing. ctDNA mutations were detected at a higher frequency in relapsed or refractory patients than in newly diagnosed patients (27.2% vs. 6.6%, respectively), authenticating the existence of spatial and genetic heterogeneity in advanced disease [159,160]. The authors also monitored tumor burden and therapeutic response through ctDNA analysis and reported that a decrease in ctDNA levels at day 5 of treatment cycle 1 correlated with superior PFS (P = 0.017). It was also concluded that ctDNA is useful for predicting disease outcomes in MM patients [161]. As mentioned earlier, MM has a precancerous state, but ctDNA detection in SMM and MGUS is still difficult [162]. In addition, as with other hematopoietic tumors, targeted gene regions have been expanded [163]. Deshpande et al. assessed whether ctDNA levels varied according to risk status defined by the 70-gene expression profile. Patients with high ctDNA levels were associated with worse PFS (hazard ratio 6.4; 95% CI 1.9–22) and overall survival rates (hazard ratio 4.4; 95% CI 1.2–15.7); ctDNA level was also elevated in the high-risk group. These findings showed that cfDNA is a dynamic tool to capture genetic events in MM [56].

Fusion and CNV of MM have been developed using WES and low-pass WGS (Guo, Manier) [164,165]. For ctDNA analysis of MM, many issues need to be resolved in the future, such as improving detection sensitivity by improving sequencing technology and prospective validation together with existing MRD measurement methods.

## 6. Future Directions and Conclusions

In 2019, Lenaerts et al. made a grand attempt to predict malignancy from copy number abnormalities by applying WGS (Genomewide Imbalance Profiling sequence) to the cfDNA of 1002 individuals with no history of malignancy. As a result, four malignant lymphoma cases and one MDS case with excess blasts were detected [166]. This announcement signifies the development from the era of cfDNA testing to the era of cfDNA-based testing. It is also true that cfDNA has its limitations. With the progress of NGS and other comprehensive genetic analysis technologies, many insights into diagnosis and treatment are moving toward clinical application. However, to expand the application of cfDNA, many issues remain to be addressed, such as the establishment of specimen processing procedures, dissemination of analytical methods that take into account cost and sensitivity, and interpretation of results in conjunction with existing testing methods. If clinicians are unaware of the benefits and limitations of cfDNA, they will not make the right decisions regarding its application to individual patients.

## Figures and Tables

**Figure 1 cancers-13-02078-f001:**
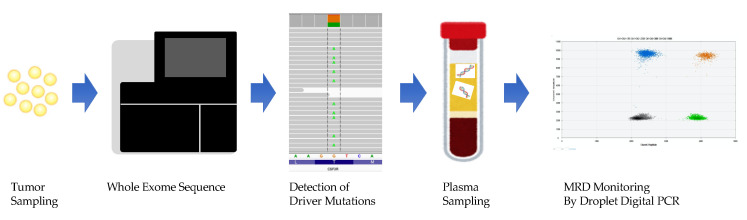
Whole-exome sequencing of bone marrow fluid before treatment to identify driver mutations and monitoring of minimal residual disease (MRD) by creating a droplet digital PCR assay for each patient.

**Table 1 cancers-13-02078-t001:** Methods of circulating tumor DNA (ctDNA) detection in peripheral blood.

Method	Sensitivity	Advantages	Limitations
qPCR (ASO-PCR) [59]	0.01%	High reproducibility of data, many facilities available for implementation	Time-consuming, laborious
Digital PCR (ddPCR) [60]	0.001%	Easy to perform, standard curve not needed	Lack of standardization in data analysis, expensive
Targeted Sequence (Panel) [61]	0.01%	Available on a commercial basis	Low sensitivity
Targeted Sequence (CAPP-seq) [62]	<0.001%	High sensitivity, comprehensive genetic analysis	Fewer facilities available
Whole Genome Sequence [63]	1%	Detect rare mutations	Suitable for tumor-rich samples

**Table 2 cancers-13-02078-t002:** Description of the studies examining cell-free DNA (cfDNA) for acute myeloid leukemia (AML)/myelodysplastic syndrome (MDS).

Reference	Diseases	N	cfDNA Material	cfDNA Isolation	cfDNA Analysis	Genes Studied	Target	Clinically Relevant Findings	New Research Perspectives
Vasioukhin, V. **1994** [70]	AML and MDS	10	Plasma in 20–30 mL PB	Phenol–chloroform method	Dot-blot screening procedure	*NRAS*	SNV	N-ras mutant genes that were not found in peripheral blood or bone marrow were detected in plasma DNA	Further studies are needed to correlate plasma DNA with peripheral blood and bone marrow DNA
Rogers, A. **2004** [71]	AML and MDS	45	Plasma	N/A	PCR and capillary electrophoresis	5q-, 7q-, +8, 17p-, 20q-, and X chromosome	LOH, X-chromosome clonality	Detection of LOH in the PB plasma of all 45 patients with cytogenetically documented chromosomal abnormalities	To test the possibility that PB can be an alternative tool to BM in assessing genetic abnormality
Gao, Y.J. **2010** [72]	AML and ALL	60	2 mL of plasma	QIAamp DNA Blood Kit (Qiagen, Hilden, Germany)	qPCR	*ACTB*	Concentrations, integrity, CNVs	The cfDNA integrity index fluctuated in correlation with the dynamics of acute leukemia	The cfDNA integrity index may be useful for MRD monitoring in acute leukemia
Iriyama, C. **2012** [73]	MDS	5	450 μL of serum	M inElute Virus Vacuum Kit (Qiagen, Hilden, Germany)	Pyrosequencing	methylation analysis (*LINE-2*), mutation detection (*TET2*)	Methylation analysis, SNV	Methylation rate decreases after azacytidine treatment in plasma DNA and the *TET2* mutant gene is detectable in BM	Testing the possibility that circulating DNA from plasma better reflects DNA from MDS clones in the BM than DNA from intact cells
Quan, J. **2015** [74]	AML	100	2 mL of plasma	QIAamp DNA Blood Kit (Qiagen, Hilden, Germany)	qPCR	*NPM*	Indel, CNVs	Copy number quantification of mutant genes established; copy number variation reflects clinical characteristics	Large-scale prospective studies to investigate the relationship between copy number of circulating NPM mutant genes and clinical outcomes are warranted
Albitar, F. **2016** [75]	MDS	16	Plasma	Nucli-SENS Easy MAG Automated Platform (BioMerieux, Marcy- l’E’ toile, France)	NGS (target sequence)	14 target genes (*ASXL1*, *ETV6*, *EZH2*, *IDH1*, *IDH2*, *NRAS*, *CBL*, *RUNX1*, *SF3B1*, *SRSF2*, *TET2*, *TP53*, *U2AF1*, *ZRSR2*)	SNV/Indel	Confirmed the presence of a neoplastic abnormal MDS clone using cfDNA NGS	Further validation in advanced MDS patients
Suzuki, Y. **2016** [76]	MDS related disease	33	Plasma and serum	QIAamp MinElute Virus Vacuum Kit or QIAamp Circulating Nucleic Acid Kit (Qiagen, Hilden, Germany)	Sanger sequence, SNP array analysis, and q-PCR or NGS (target sequence)	Sanger, SNP array, and q-PCR (*IDH2*, *SETBP1*, *U2AF1*, *SRSF2*, *NRAS*, *TET2*, and *FLT3*), NGS (39 targeted genes)	SNV/Indel	Detection of cfDNA that varies in correlation with disease state	Verification of the possibility that cfDNA reflects the disease status of MDS
Christenson, E.S. **2017** [90]	AML, MDS	7	Plasma in 10 mL PB	QIAamp Circulating Nucleic Acid Kit (Qiagen, Hilden, Germany)	ddPCR	*SF3B1*	SNV	Reported that SNPs in wild-type alleles affect allele frequencies detected by ddPCR	Validation of a companion diagnostic method using ctDNA ddPCR
Yeh, P. **2017** [96]	MDS	12	Plasma	QIAamp Circulating Nucleic Acid Kit (Qiagen, Hilden, Germany)	NGS (target sequence) and/or ddPCR	55 target genes	SNV/Indel	Prediction of treatment failure by tracking driver mutations and karyotype abnormalities using ctDNA and evidence that ctDNA dynamics reflects tumor burden in MDS	Application of ctDNA analysis as a non-invasive biomarker to complement existing monitoring strategies for MDS
Nakamura, S. **2018** [86]	AML, MDSMM, NHL	17	Serum	QIAamp Circulating Nucleic Acid Kit (Qiagen, Hilden, Germany)	ddPCR	Genes whose mutations were detected by WES (*STAG2*, *JAK3*, *NRAS*, *KRAS*, *TP53*, *DNMT3A*, *NPM1*, *GATA2*, *MYD88*, *B2M*, *SF3B1*, *U2AF1*)	SNV/Indel	ctDNA monitoring facilitated the identification of patients with hematological cancers at risk of recurrence prior to established clinical parameters	Need to implement ctDNA monitoring of hematopoietic tumor patients on an even larger scale
Zhong, L. **2018** [89]	AML	235	Plasma	QIAamp DNA Mini Kit (Qiagen GmbH, Hilden, Germany)	PCR and gel electrophoresis, qPCR	*IGH* and *TCR*	Ig-gene and TCRγ rearrangement	Detection of monoclonal IGH and TCR rearrangements in AML ctDNA	Application of monoclonal IGH and TCR rearrangements for MRD assessment in AML
Zhao, P. **2019** [77]	MDS	26	Plasma	N/A	NGS (target sequence)	127 target genes	SNV/Indel	ctDNA reflects genetic variation in BM DNA and is useful for monitoring the pathogenesis of MDS	Application of ctDNA for prognosis prediction of MDS
Nakamura, S. **2019** [87]	AML/MDS	53	Serum	QIAamp Circulating Nucleic Acid Kit (Qiagen, Hilden, Germany)	ddPCR	Genes whose mutations were detected by WES (*DNMT3A*, *STAG2*, *SRSF2*, *SF3B1*, *WT1*, *GATA2*, *NPM1*, *CEBPA*, *IDH1*, *IDH2*, *TP53*, *U2AF1*, *BCORL1*, *ATRX*, *ASXL1*, *RUNX1*, *CEBPA*, *SH2B3*, *KIT*, *PTPN11*, *ETV6*, *RAD21*, *CSF3R*, *CTCF*, *ETNK1*, *KMT2D*, *BCOR*, *XPO7*)	SNV/Indel	Prediction of relapse by MRD monitoring using ddPCR combined with NGS after hematopoietic stem cell transplantation	Conducting prospective tests
Short, N.J. **2020** [88]	AML	22	Plasma in 10 mL PB	QIAamp Circulating Nucleic Acid Kit (Qiagen, Hilden, Germany)	NGS (target sequence)	275 target genes	SNV/Indel	Detection of residual lesions in cfDNA specimens in remission by targeted sequencing	Evaluation the prognostic impact of MRD as detected by ctDNA sequencing
Zeidan, A.M. **2020** [95]	AML	20	Plasma	QIAamp Circulating Nucleic Acid Kit (Qiagen, Hilden, Germany)	ddPCR	Genes whose mutations were detected by target sequence	SNV/Indel	Monitoring of ctDNA in a Phase Ib study of the PLK1 inhibitor, onvansertib, showed tumor burden during therapy	Evaluating the utility of serial ctDNA measurements as a predictor of clinical response

BM, bone marrow; CNV, copy number variation; ddPCR, droplet digital PCR; LOH, loss of heterozygosity; MM, multiple myeloma; NGS, next-generation sequence; NHL, non-Hodgkin lymphoma; PB, peripheral blood; qPCR, quantitative PCR; SNP, single-nucleotide polymorphism; SNV, single-nucleotide variants; WES, whole-exome sequence.

## Data Availability

Not applicable.

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
