# Peer review of "Role of Circulating Tumor DNA in Hematological Malignancy"

_cancers, 2021, doi:10.3390/cancers13092078_

Round 1
Reviewer 1 Report
Authors have fully replied to all criticism. I have no other questions to raise
Reviewer 2 Report
The authors addressed all the issues raised during the first revision round. I have no other additional comments
Reviewer 3 Report
The authors have done a good job reorganizing their manuscript following the reviewers' comments.
The inclusion of appropriate references, the tables that summarize key points, as well as the revision of significant parts of the manuscript make it easier to read and a useful literature review.
This manuscript is a resubmission of an earlier submission. The following is a list of the peer review reports and author responses from that submission.
Round 1
Reviewer 1 Report
The work of Ogawa et al. it is very interesting because it reviews the current literature on the topic of liquid biopsies. The contents of the text are and the references are adequate. The part of the statistics is not evaluable. My suggestion is to insert a summary table of the main results published on the subject of haemopathies, dividing it into: 1) pathology analyzed 2) type of sample analyzed 3) type of transcript / mutation sought 4) results 5) new research perspectives
All acronyms must be specified in the text or in a table
Reviewer 2 Report
The present manuscript provides an overview of the role of ctDNA analyses in hematological malignancy, focusing on feasibility, limitations, and areas of clinical application.
Some comments:
- Table 1 should be revised, as in the present format is not clearly readable.
- Clonal hematopoiesis is an emerging clinical issue in the liquid biopsy field [Rolfo C, et al. Crit Rev Oncol Hematol 2020]. The identification of non-tumor derived mutations in cfDNA might be associated with false positive results, especially in the context of early detection and minimal residual disease monitoring. This topic and the strategies to limit it should be discussed [see for instance, Chan HT, et al. Cancers 2020].
- A table or figure describing the different methodologies with advantages and disadvantages (limit of detection, costs, turnaround time, etc.) would be useful.
- The use of the term “exosomes” should be replaced with the more appropriate “extracellular vesicles” [Thery C, et al. J Extracell Vesicles. 2018]
- Among the different components of the large liquid biopsy field, microRNAs (miRNAs) have been extensively studied also in hematological malignancies. A brief paragraph summarizing the most relevant studies/results in the field would be useful [Papanota AM, et al. Int J Mol Sci. 2021; Han Z, et al. Curr Opin Oncol. 2020]
Reviewer 3 Report
The present manuscript is a review entitled "Role of ctDNA in hematological malignancy", however, this reviewer does not feel that the content justifies the title.
Almost half of the manuscript is a not so successful attempt to give an overview of liquid biopsies in general. However, these sections omit critical information about the use of different liquid biopsy biomarkers, such as the CTCs and the exosomes. Especially, in the exosomes section there is very little about their clinical applications (in what types of cancer, clinical value, methods of detection, landmark studies etc) and more about their role in tumor biology in general.
In the ctDNA section:
- ref.39 is outdated; mor recent publications report cfDNA levels up to 100 ng/ml in healthy individuals.
- why "ctDNA and disease burden" is a separate subsection?
- "application to solid cancers" is missing landmark studies
- the "quality control" sub-section includes data generated only from one group and it is not in agreement with the widely accepted notions about pre-processing of ctDNA. Furthermore, quality control in liquid biopsies does not have to do only with the amount of ctDNA.
In my opinion, the introductory part about liquid biopsies should be omitted, since the title of the manuscript refers to the use of ctDNA and not liquid biopsies in general.
The authors should only focus on hematological malignancies, what their particularities are compared to solid tumors, and how ctDNA could provide new crucial information that otherwise wouldn't be available.